# Determinants of gestational weight gain during pregnancy in a multiethnic UK-based population: Findings from the Born in Bradford cohort study

Petra A. T. Araújo[1], Maria A. Quigley[1], Gillian Santorelli[2], Victoria Coathup[1]*

1 National Perinatal Epidemiology Unit, Nuffield Department of Population Health, University of Oxford, Oxford, Oxfordshire, United Kingdom, 2 Bradford Institute for Health Research, Bradford Royal Infirmary, Bradford, United Kingdom

* victoria.coathup@npeu.ox.ac.uk

## Abstract

### Introduction

Unhealthy maternal weight gain during pregnancy is associated with deleterious outcomes to mothers and their offspring. Current literature on the determinants of gestational weight gain yields inconsistent results, with limited research conducted in the United Kingdom. This study investigates potential determinants of unhealthy gestational weight gain in a multiethnic cohort within Bradford, United Kingdom.

### Methods

The study analysed 7,769 singleton pregnancies from the Born in Bradford Cohort. Women were enrolled at ~26 weeks' gestation. Weight at first antenatal appointment, recruitment and/or third trimester were used to calculate weekly average weight gain. This was categorized as 'less than recommended', 'recommended' or 'more than recommended' based on the Institute of Medicine (IOM) criteria. Associations between potential determinants and gestational weight gain were assessed using multinomial logistic regression with recommended gestational weight gain as the reference.

### Results

Overall, 22.4% of women gained weight within the recommended range; 20.3% gained less than recommended, and 57.3% gained more than recommended. Key risk factors for gaining less weight than recommended were unhealthy baseline BMI (aOR=1.78 for underweight, aOR=1.3 for obese), higher parity (e.g. aOR=1.46 for 3 + children) and lower socioeconomic status (aOR=1.4). The strongest risk factors for gaining more weight than recommended were high baseline BMI (e.g. aOR=5.86 for obese) and higher psychiatric morbidity score (aOR=1.22); being underweight (aOR=0.58) and higher parity (e.g. aOR=0.70 for 3 + children) were associated with

**Data availability statement:** Data can be accessed via Born in Bradford once a proposal has been submitted and approved. For more information regarding the data access process, please see: https://borninbradford.nhs.uk/research/how-to-access-data/

**Funding:** This research was funded by an Intermediate Research Fellowship from the Nuffield Department of Population Health at the University of Oxford, UK (H6D00010). The fellowship was awarded to Victoria Coathup. The funders had no role in study design, data collection and analysis, decision to publish, or preparation of the manuscript.

**Competing interests:** The authors have declared that no competing interests exist.

a lower risk of gaining more weight than recommended. The effect of mental health seemed to be particularly important among women of Pakistani background, while parity seemed to play a major role among White British women.

## Conclusion

Baseline BMI, age, socioeconomic position, parity and mental health are associated with unhealthy weight gain during pregnancy in a multiethnic UK population. These findings can help identify at-risk women and inform targeted preventative strategies.

## Introduction

Maternal weight gain during pregnancy supports the increased metabolic demands associated with gestation [1]. To optimize pregnancy outcomes, maternal weight gain should be kept within a healthy range [2]. Numerous studies suggest that gestational weight gain (GWG) below and above recommendations may be associated with short and long-term deleterious consequences to mothers and their offspring [2–6]. For example, gaining more weight than recommended has been linked with gestational diabetes [2], preeclampsia [2], large-for-gestational-age neonates [2,4], macrosomia [4], higher rates of cesarean delivery [4], childhood obesity [5–7] and impaired cognitive skills in children [6,8]. Meanwhile, gaining less weight than recommended is associated with small-for-gestational-age neonates [2,4] and prematurity [2,4].

The importance of attaining a healthy weight gain during pregnancy has been increasingly recognised in recent decades. Nevertheless, the number of women with GWG below or above the recommended 2009 Institute of Medicine (IOM) limits remains remarkably high [4,6]. For example, a systematic review and meta-analysis published in 2017 estimated that the overall prevalence of weight gain within the recommended range during pregnancy was only 30%, while 23% of women gained less weight than recommended, and 47% gained more weight than recommended [4].

Considering the potential detrimental impact of unhealthy GWG, it is imperative to identify which risk factors are associated with 'less than recommended' and/or 'more than recommended' maternal weight gain. Previous studies have attempted to pinpoint these determinants, and various potential risk factors have been suggested, spanning physical, psychological and socio-economic characteristics [9–12]. Nevertheless, findings are not consistent across the literature. This heterogeneity may be partially explained by differences in population characteristics and/or different definitions and measurements of GWG.

Despite the global efforts to identify such determinants, there is a scarcity of data from the United Kingdom (UK) [13], where only four other studies [13–16] have addressed this topic – two of them with small sample sizes. Additionally, most of the international literature focuses on determinants of GWG above the reference ranges, while the determinants of GWG below the reference ranges are under-explored.

The present study aims to overcome these deficits by investigating the potential determinants of GWG below and above IOM recommendations in an ethnically

diverse cohort in the city of Bradford (England, United Kingdom). The main objective was to examine the potential impact of a wide range of factors – including demographic characteristics, socio-economic status, physical and mental health, anthropometric measurements, ethnicity, lifestyle and reproductive history – on unhealthy GWG.

By contributing to the literature on this topic, this study could potentially help inform evidence-based health promotion strategies, as well as facilitate the identification of groups at risk and ensure better surveillance, follow-up and counselling to pregnant women and their families.

## Methods

### Setting and study design

The present study utilises data from the Born in Bradford (BiB) birth cohort study. This is a longitudinal research programme established in a large and economically deprived city in the north of England, aiming to investigate diverse aspects of maternal and child health. The population of Bradford is predominantly comprised of young individuals from diverse ethnic backgrounds (mainly White-British and Pakistani) [17]. We conducted a secondary analysis of data from this birth cohort study.

All pregnant women who received antenatal care in the city of Bradford from 2007 to 2011 and planned to give birth at the Bradford Royal Infirmary (the only maternity unit in the city) were invited to participate. Most women were typically enrolled at 26–28 weeks of gestational age when they attended an 'oral glucose tolerance test' (oral GTT) appointment. The oral GTT was offered to all pregnant women booked for delivery at the Bradford Royal Infirmary, with an uptake of approximately 80% [18].

At the time of enrolment, participants completed an interviewer-administered questionnaire (administered mainly in English, Urdu or Mirpuri) and had their weight and height measured and recorded. They were then followed-up through linkage to maternity notes. Further details on enrolment and follow-up of participants have been extensively described elsewhere [17,18].

### Eligibility criteria

The present analysis included women with singleton pregnancies and live, term births (≥37 to ≤42 weeks' gestation at birth). Participants were excluded if they experienced hyperemesis gravidarum or if their GWG could not be adequately calculated. This encompassed women who initiated antenatal care after 14 weeks gestation, those who lacked information on baseline BMI and those who had less than two weight measurements during pregnancy. Additionally, participants were excluded if the interval between the initial and final weight measurement was less than eight weeks, if information on gestational age at the time of measurement was lacking, or if weight gain was implausible (±4SD from the mean weight gain standardised by gestational age at final weight measurement). Participants who moved outside of Bradford shortly after giving birth and those whose infants died on the day of birth were also excluded from the analyses (S1 Fig).

### Outcome

Maternal weight was measured at three times points during pregnancy: (1) during the first antenatal appointment by a midwife; (2) along with height, at study recruitment by a research nurse (~26–28 weeks' gestation); and (3) during a routine antenatal appointment in the third trimester by a midwife. Median gestational age at the first and third trimester antenatal appointments were 11 weeks (IQR 10–12 weeks) and 36 weeks (IRQ 35–37 weeks), respectively. Average weekly weight gain was calculated as follows: (final weight - weight at first antenatal appointment)/ number of weeks between the two time points. For those without a third trimester measurement, weight at recruitment was used as the "final weight" for this calculation. The median gestational age at final weight measurement was 35 weeks (IQR 26–36). The number of weeks between initial and final weight measurements varied from 8 to 33 weeks, with a median of 23 (IQR 16–26).

The main outcome consisted of a three-level categorical variable representing the adequacy of GWG: 'less than the recommended weight gain (RWG)', 'recommended weight gain (RWG)' and 'more than the recommended weight gain (RWG)'. The Institute of Medicine (IOM) 2009 [19] criteria were used to categorize participants according to their average weekly weight gain. Results were then compared to IOM thresholds for recommended weekly GWG throughout the second and third trimesters, which vary according to baseline BMI, as follows – underweight: 0.44 to 0.58 kg/week; healthy weight: 0.35 to 0.50 kg/week; overweight: 0.23 to 0.33 kg/week; and obese: 0.17 to 0.27 kg/week (S1 Table). Women's weight measured during their first antenatal visit (~11 weeks' gestation) was used to calculate baseline BMI. Considering that weight gain during the first trimester is usually small (0.5 to 2 kg in total) [19], early pregnancy weight was used as a proxy of pre-pregnancy weight. Different BMI cutoffs were used according to participants' ethnic background (S2 and S3 Tables). For women of south-Asian ethnic origin, the thresholds were: underweight <18.5 kg/m², normal weight 18.5–22.9 kg/m², overweight 23.0–24.9 kg/m², obese ≥25 kg/m² [20]. For other participants (including White-British women), the thresholds were <18.5 kg/m², 18.5–24.9 kg/m², 25–29.9 kg/m² and ≥ 30 kg/m², respectively.

## Potential determinants

Twelve potential determinants of GWG below or above IOM recommendations were selected based on a literature search, their *a priori* plausibility, and on the availability of data within the BiB cohort:

**Demographic information.** 1) Maternal age at birth: <20 years, 20–24, 25–29, 30–34 and ≥35 years; 2) Ethnicity: White British, Pakistani and 'other';

**Socioeconomic information.** 3) Socioeconomic position (SEP): Based on a published latent-class analysis of 19 different items [21], participants were classified into five SEP groups: "least socioeconomically deprived and most educated"; "employed and not materially deprived"; "employed and no access to money"; "benefits but coping" and "most economically deprived"; 4) Quintiles of the index of multiple deprivation (IMD), a marker of neighbourhood-level disadvantage used in the UK [22], with the 1st quintile being the most deprived; 5) Education: the highest level of attained maternal education (equivalised for UK qualifications): <5 GCSE (where GCSE examinations are typically taken at the end of mandatory education ~16 years); 5 GCSE (where 5 is a common prerequisite for A level study); A-level (education level typically completed by ~18 years); >A-level (higher education typically completed beyond 18 years); and other/unknown; 6) Marital/cohabitation status: "married and living with a partner", "not married and living with a partner", and "not living with a partner"; 7) Current employment (yes and no);

**Lifestyle, obstetric history, and health.** 8) Smoking during pregnancy (yes and no); 9) Parity: the number of previous births were categorized into four groups: nulliparous, 1, 2 and 3+ births; 10) Baseline BMI (kg/m²) categorized into underweight, healthy weight, overweight and obese (see S2 Table for category definitions); 11) Previous hypertension (yes/no); 12) Mental health: based on responses to the General Health Questionnaire-28 (GHQ-28) [23], administered at recruitment, patients were classified as "at higher risk for psychiatric morbidity" vs "at lower risk for psychiatric morbidity". The binary variable was derived from the total scores using group-specific 75th centiles as a threshold. These cut-offs have been personalized for different combinations of ethnic background and language of questionnaire administration in a previous BiB study [24]. Further information on the GHQ-28 binary variable is provided in the supplementary information (section S1, S2 in S1 File).

## Statistical analyses

Data were searched for missing values and inconsistent/implausible observations. Counts and proportions were calculated for binary and categorical variables. The distribution of continuous variables was assessed through histograms and quantile-quantile plots. Means and standard deviations were computed for variables with normal distribution, and median and interquartile ranges (IQR) for variables with skewed distributions. After the completion of the exploratory analyses, chi-square tests and ANOVA were performed to assess whether differences in baseline characteristics across GWG

groups were statistically significant. Crude multinomial logistic regression models were used to investigate the unadjusted associations between each potential determinant and GWG categories. Odds ratios and corresponding 95% confidence intervals for 'less than the RWG' and for 'more than the RWG' (compared to RWG) were estimated for each level of the determinants. Variables associated with the outcome (p < 0.1 Wald test) were selected to be included in multivariable models. An *a priori* decision to include age and ethnicity in the multivariable models, irrespective of the results of the crude analyses, was made due to the theoretical importance of these variables for pregnancy outcomes and for the BiB cohort, respectively.

Initially, multivariable models including socioeconomic characteristics (and additionally adjusted for age) were fitted to investigate which of them would be independently associated with GWG once mutually controlled for each other. The five SES variables were added sequentially based on the strength of univariable associations. Once a final model was achieved, only those displaying a significant association (p < 0.05) in at least one of the individual Wald tests were selected to be included in the next step of model building. At this point, the "lifestyle, obstetric history, and health" variables were added sequentially using the same criteria and approach previously described. Once a "full model" was achieved, only variables that remained significant at the < 0.05 level in at least one of the individual categories were kept in the final model (in addition to age and ethnicity).

**Subgroup analysis.** Considering the relevance of ethnicity within Bradford and previously reported differences in maternal weight, height, and fetal weight between Pakistani and White British women within the BiB cohort [17], an exploratory subgroup analysis, stratifying the final model by ethnic group, was deemed appropriate. Considering the exploratory/preliminary nature of this analysis and aiming to avoid multiple testing, a decision was made not to perform formal tests for interaction.

**Missing data.** A final GHQ-28 score was not attainable for 2,900 pregnancies (22.17% of the total sample). There was less than 10% of missing data for other variables. Patterns of missingness were evaluated by building crude and multivariable logistic regression models with the availability vs non-availability of data on GHQ-28 as the outcome. The pattern of missingness was deemed to be "missing at random". Multiple imputation by chained equation (MICE) with 20 imputations was used, and results were combined using Rubin's rules [25]. All potential determinants, the outcome (categories of gestational weight gain), and two auxiliary variables (gestational age at first antenatal appointment and gestational age at birth) were included in the imputation model. The main multivariable analyses reported in this study are based on the imputed models.

## Sensitivity analyses

A series of sensitivity analyses were performed to assess the robustness of the final model against limitations and potential biases: 1) excluding participants with gestational diabetes mellitus; 2) restricting the analysis to post-recruitment weight gain (weight gain between recruitment and third trimester measurement); 3) using categories of absolute (total) weight gain over pregnancy as the outcome; 4) using standard ('Western') BMI cut-offs for all participants, irrespective of ethnic background; 5) adding 0.5 and then 2 kg to weight at first antenatal appointment and fixing gestational age at booking at 14 weeks for participants who had a first antenatal visit before 8 weeks; 6) a complete case analysis (CCA) was performed as a sensitivity analysis. Due to the high level of missing GHQ-28 data, the CCA models did not include an assessment of mental health as a potential determinant. Further information on sensitivity analyses can be found in the supplementary information (S2 Fig).

Regression models and chi-square tests were clustered for participants' identification number, since a subset of women had more than one pregnancy within the BiB cohort. All statistical analyses were undertaken using Stata/MP versions 17.0 and 18.0 (StataCorp).

**Ethical considerations.** Born in Bradford is a family of research studies including three longitudinal multi-ethnic birth cohorts (Born in Bradford; Born in Bradford's Better Start and BiB4All). Ethical approval for all aspects of the research was

granted by Bradford Research Ethics Committee (Ref 07/H1302/112, 15/YH/0455 and 17/YH/0202). Participants signed a written informed consent for data collection and linkage to routine healthcare records upon enrollment to the study. Approval to conduct a secondary analysis of Born in Bradford data was granted by Bradford Institute for Health Research on 6th November 2019. All Born in Bradford survey data was de-identified.

## Results

The initial dataset encompassed 13,858 pregnancies (12,453 women). After the exclusion of 6,089 observations that did not meet the inclusion criteria, a total of 7,769 pregnancies (7,335 women) remained in the analysis (S1 Fig). Table 1 shows baseline characteristics overall and by GWG group. Participants' ages ranged from 15 to 49 years (mean 27.7, SD 5.5). The number of White British and Pakistani women was similar, and they represented 41.0% and 44.6% of the study sample, respectively. The most common level of educational attainment was 5 GCSE equivalent (31.7%) and more than half of women (53.8%) were currently not employed. Thirty-six percent of participants lived in the most socio-economically deprived neighbourhoods (IMD quintile 1), while only 3.6% lived in one with an IMD within the fifth quintile. Median baseline BMI was 24.9 kg/m$^2$ (IQR 21.9–29.0) and 38.1% of women had a healthy BMI at baseline, whereas 4.3% were underweight, 31.1% overweight, and 26.5% obese.

Among study participants, 58.3% had information on third trimester weight (≥28 weeks' gestation), with 42.1% having a late third trimester weight measurement (≥36 weeks' gestation). Overall, less than one quarter of women (22.4%) gained weight within the recommended range, while 20.3% gained below the recommended range and 57.3% gained above the recommended range. All baseline characteristics, except ethnicity and hypertension, were significantly associated with GWG (Table 1).

### GWG below the recommended range

Women with an unhealthy pre-pregnancy BMI (either underweight, overweight or obese), higher levels of economic deprivation, multiparity and those who smoked during pregnancy had higher odds of gaining less than the RWG in univariable analyses (Table 2). Higher levels of education were associated with decreased odds of gaining less than the RWG. No associations with the other variables were identified.

In the final multivariable model (Fig 1a), BMI, age, socioeconomic position and parity were associated with GWG below the recommended range. Underweight, overweight and obese women were at greater risk of gaining less than the RWG when compared to individuals with a healthy baseline BMI [OR=1.78 (95% CI 1.36–2.32, p<0.001); OR=1.37 (95% CI 1.16–1.63, p<0.001); OR=1.30 (1.05–1.60, p=0.014), respectively]. Women in worse socioeconomic positions also faced increased odds of gaining less weight than the recommended threshold. Those classified under "benefits but coping" had 42% higher odds of having GWG below IOM guidelines when compared to least deprived participants [OR=1.42 (95% CI 1.14–1.77, p=0.002)]. Similarly, those in the "most deprived" category had 37% higher odds of experiencing low GWG compared to the same reference group [OR=1.37 (95% CI 1.07–1.76, p=0.014)]. All categories of parity were associated with increased odds of gaining less weight than recommended, ranging from 23% [OR=1.23 (95% CI 1.03–1.46, p=0.022)] for women who had given birth once before to 46% [OR=1.46 (95% CI 1.11–1.92, p=0.007)] for those who had previously given birth three or more times, when compared to nulliparous women. Finally, participants aged 20–24 years had 26% higher odds of experiencing lower than recommended weight gain when compared to those aged 25–29 years [OR 1.26 (95% CI 1.01–1.47, p=0.043)].

### GWG above the recommended range

Baseline BMI, parity, mental health, age, IMD and marital status influenced the odds of gaining more than the RWG in univariable analyses (Table 2). In the final multivariable model (Fig 1b), baseline BMI played a major role in predicting higher-than-recommended weight gain during pregnancy. While being underweight led to a lower risk of experiencing GWG

**Table 1. Baseline characteristics of study participants by category of gestational weight gain.**

| | Total | Gestational weight gain | | | p-value* |
|---|---|---|---|---|---|
| | | Less than recommended | Recommended | More than recommended | |
| | n = 7,769 | n = 1,577 | n = 1,743 | n = 4,449 | |
| **Maternal age** (categorical, years) | | | | | <0.001 |
| <20 | 430 (5.5%) | 107 (6.8%) | 115 (6.6%) | 208 (4.7%) | |
| 20-24 | 1,981 (25.5%) | 452 (28.7%) | 454 (26.0%) | 1,075 (24.2%) | |
| 25-29 | 2,562 (33.0%) | 499 (31.6%) | 582 (33.4%) | 1,481 (33.3%) | |
| 30-34 | 1,821 (23.4%) | 326 (20.7%) | 398 (22.8%) | 1,097 (24.7%) | |
| 35+ | 975 (12.5%) | 193 (12.2%) | 194 (11.1%) | 588 (13.2%) | |
| Missing | 0 | 0 | 0 | 0 | |
| **Ethnicity** | | | | | 0.243 |
| White British | 3,180 (41.0%) | 654 (41.5%) | 716 (41.2%) | 1,810 (40.7%) | |
| Pakistani | 3,464 (44.6%) | 675 (42.8%) | 766 (44.1%) | 2,023 (45.5%) | |
| Other | 1,117 (14.4%) | 248 (15.7%) | 256 (14.7%) | 613 (13.8%) | |
| Missing | 8 | 0 | 5 | 3 | |
| **Mental Health**** | | | | | 0.013 |
| GHQ-28 < 75th centile | 1,397 (22.6%) | 269 (21.7%) | 272 (20.1%) | 856 (23.8%) | |
| GHQ-28 > 75th centile | 4,786 (77.4%) | 969 (78.3%) | 1,083 (79.9%) | 2,734 (76.2%) | |
| Missing | 1,586 | 339 | 388 | 859 | |
| **Education** | | | | | <0.001 |
| <5 GCSE | 1,574 (20.6%) | 378 (24.4%) | 335 (20.0%) | 861 (19.6%) | |
| 5 GCSE | 2,420 (31.7%) | 541 (34.9%) | 533 (31.8%) | 1,346 (30.6%) | |
| A-level | 1,136 (14.9%) | 201 (13.0%) | 245 (14.6%) | 690 (15.7%) | |
| Higher than A-level | 2,003 (26.3%) | 334 (21.5%) | 470 (28.0%) | 1,199 (27.3%) | |
| Other/unknown | 491 (6.4%) | 96 (6.2%) | 94 (5.6%) | 301 (6.8%) | |
| Missing | 145 | 27 | 66 | 52 | |
| **Currently employed** | | | | | <0.001 |
| No | 4,149 (53.8%) | 918 (58.3%) | 947 (55.7%) | 2,284 (51.4%) | |
| Yes | 3,569 (46.2%) | 656 (41.7%) | 752 (44.3%) | 2,161 (48.6%) | |
| Missing | 51 | 3 | 44 | 4 | |
| **SEP†** | | | | | <0.001 |
| Least deprived | 1,542 (20.1%) | 255 (16.2%) | 364 (21.6%) | 923 (20.9%) | |
| Employed not mat. depr. | 1,616 (21.0%) | 286 (18.2%) | 324 (19.2%) | 1,006 (22.7%) | |
| Emp. no access to money | 1,176 (15.3%) | 240 (15.3%) | 281 (16.7%) | 655 (14.8%) | |
| Benefits but coping | 2,187 (28.5%) | 511 (32.5%) | 471 (27.9%) | 1,205 (27.2%) | |
| Most deprived | 1,161 (15.1%) | 278 (17.7%) | 247 (14.6%) | 636 (14.4%) | |
| Missing | 87 | 7 | 56 | 24 | |
| **Marital status** | | | | | <0.001 |
| Married and living with partner | 5,162 (67.0%) | 995 (63.2%) | 1,108 (65.4%) | 3,059 (68.9%) | |
| Not married and living with partner | 1,392 (18.1%) | 290 (18.4%) | 314 (18.5%) | 788 (17.7%) | |
| Not living with partner | 1,154 (15.0%) | 289 (18.4%) | 271 (16.0%) | 594 (13.4%) | |
| Missing | 61 | 3 | 50 | 8 | |
| **IMD Quintiles**** | | | | | <0.001 |
| 1 | 2,802 (36.3%) | 610 (38.7%) | 611 (36.0%) | 1,581 (35.6%) | |
| 2 | 2,081 (27.0%) | 449 (28.5%) | 461 (27.2%) | 1,171 (26.3%) | |
| 3 | 1,497 (19.4%) | 285 (18.1%) | 333 (19.6%) | 879 (19.8%) | |

*(Continued)*

**Table 1.** (Continued)

| | Total | Gestational weight gain | | | p-value* |
| --- | --- | --- | --- | --- | --- |
| | | Less than recommended | Recommended | More than recommended | |
| | n = 7,769 | n = 1,577 | n = 1,743 | n = 4,449 | |
| 4 | 1,065 (13.8%) | 201 (12.8%) | 240 (14.1%) | 624 (14.0%) | |
| 5 | 275 (3.6%) | 31 (2.0%) | 52 (3.1%) | 192 (4.3%) | |
| Missing | 49 | 1 | 46 | 2 | |
| **Parity** | | | | | <0.001 |
| Nulliparous | 3,071 (41.2%) | 547 (35.9%) | 696 (41.5%) | 1,828 (42.9%) | |
| 1 | 2,176 (29.2%) | 483 (31.7%) | 515 (30.7%) | 1,178 (27.7%) | |
| 2 | 1,258 (16.9%) | 291 (19.1%) | 288 (17.2%) | 679 (16.0%) | |
| 3+ | 953 (12.8%) | 201 (13.2%) | 180 (10.7%) | 572 (13.4%) | |
| Missing | 311 | 55 | 64 | 192 | |
| **Smoked during pregnancy** | | | | | <0.001 |
| Yes | 1,214 (15.7%) | 301 (19.1%) | 274 (16.2%) | 639 (14.4%) | |
| No | 6,502 (84.3%) | 1,274 (80.9%) | 1,422 (83.8%) | 3,806 (85.6%) | |
| Missing | 53 | 2 | 47 | 4 | |
| **Previous hypertension** | | | | | 0.498 |
| Yes | 62 (0.8%) | 11 (0.7%) | 11 (0.6%) | 40 (0.9%) | |
| No | 7,458 (99.2%) | 1,513 (99.3%) | 1,684 (99.4%) | 4,261 (99.1%) | |
| Missing | 249 | 53 | 48 | 148 | |
| **Baseline BMI (kg/m²)** | 24.9 (21.9-29.0) | 23.3 (21.0-27.1) | 22.9 (20.7-26.0) | 26.3 (23.2-30.9) | <0.001 |
| **Baseline BMI†† (categorical)** | | | | | <0.001 |
| Underweight | 337 (4.3%) | 148 (9.4%) | 110 (6.3%) | 79 (1.8%) | |
| Healthy weight | 2,957 (38.1%) | 754 (47.8%) | 992 (56.9%) | 1,211 (27.2%) | |
| Overweight | 2,413 (31.1%) | 418 (26.5%) | 396 (22.7%) | 1,599 (35.9%) | |
| Obese | 2,062 (26.5%) | 257 (16.3%) | 245 (14.1%) | 1,560 (35.1%) | |
| Missing | 0 | 0 | 0 | 0 | |

*P- values refer to the results of ANOVA and chi-square tests clustered for mother's identification number.

**GHQ-28: General health questionnaire. > 75th centile means higher risk of psychiatric morbidity.

***IMD: index of multiple deprivation.

†Categories of SEP (socioeconomic position): "least socioeconomically deprived and most educated"; "employed and not materially deprived"; "employed and no access to money"; "benefits but coping" and "most economically deprived"

††BMI (body mass index) categories' thresholds vary by ethnicity (S2 Table).

above IOM guidelines [OR=0.58 (95% CI 0.43–0.79, p<0.001)]; being overweight or obese dramatically increased the odds [OR=3.56 (95% CI 3.09–4.10, p<0.001) and 5.86 (95% CI 4.96–6.92, p<0.001), respectively]. One of the categories of SEP ('employed and no access to money') was associated with lower odds of gaining more than the RWG when compared to least deprived individuals. All categories of multiparity were associated with lower odds of acquiring more than the recommended weight when compared to nulliparity. Pregnancies with a higher risk of psychiatric morbidity (GHQ-28>75th centile) were associated with higher odds of gaining weight above recommendations [OR=1.22 (95% CI 1.07–1.39, p=0.003].

## Subgroup analysis: Maternal ethnicity

Results for the subgroup analyses are shown on Fig 2. The effect of high multiparity (3+previous births) on the odds of having a GWG below the recommended range seemed to be especially relevant among White British women. The impact

**Table 2. Crude odds ratios and 95% confidence intervals for 'less than the recommended weight gain' and 'more than the recommended weight gain' by each potential determinant.**

| Potential determinants | Less than the RWG | | | More than the RWG | | |
|---|---|---|---|---|---|---|
| | OR | 95% CI | p-value* | OR | 95% CI | p-value* |
| **Maternal age (years)** | | | | | | |
| <20 | 1.09 | 0.81-1.45 | 0.580 | **0.71** | **0.55-0.91** | **0.007** |
| 20-24 | **1.16** | **0.97-1.39** | **0.098** | 0.93 | 0.80-1.08 | 0.335 |
| 25-29 | Ref. | – | – | Ref. | – | – |
| 30-34 | 0.96 | 0.79-1.16 | 0.637 | 1.08 | 0.93-1.29 | 0.296 |
| 35+ | 1.16 | 0.92-1.46 | 0.207 | **1.19** | **0.99-1.44** | **0.071** |
| **Ethnicity** | | | | | | |
| White British | Ref. | – | – | Ref. | – | – |
| Pakistani | 0.96 | 0.83-1.12 | 0.636 | 1.04 | 0.93-1.18 | 0.479 |
| Other | 1.06 | 0.86 -1.30 | 0.573 | 0.95 | 0.80-1.12 | 0.536 |
| **Mental Health**** | | | | | | |
| GHQ-28 < 75th centile | Ref. | – | – | Ref. | – | – |
| GHQ-28 > 75th centile | 1.11 | 0.91-1.34 | 0.300 | **1.25** | **1.07-1.45** | **0.005** |
| **Education** | | | | | | |
| <5 GCSE | Ref. | – | – | Ref. | – | – |
| 5 GCSE | 0.90 | 0.74-1.09 | 0.276 | 0.98 | 0.84-1.16 | 0.832 |
| A-level | **0.73** | **0.57-0.92** | **0.009** | 1.10 | 0.90-1.33 | 0.357 |
| Higher than A-level | **0.63** | **0.51-0.77** | **<0.001** | 0.99 | 0.84-1.17 | 0.930 |
| Other/unknown | 0.91 | 0.66-1.25 | 0.544 | 1.25 | 0.96-1.62 | 0.101 |
| **Currently employed** | | | | | | |
| No | Ref. | – | – | Ref. | – | – |
| Yes | 0.90 | 0.78-1.03 | 0.135 | **1.19** | **1.06-1.33** | **0.002** |
| **SEP**† | | | | | | |
| Least deprived | Ref. | – | – | Ref. | – | – |
| Employed not mat. dep. | **1.26** | **1.00 -1.58** | **0.046** | **1.22** | **1.03-1.46** | **0.024** |
| Employed no access to money | 1.22 | 0.96-1.54 | 0.100 | 0.92 | 0.76-1.11 | 0.374 |
| Benefits but coping | **1.55** | **1.26-1.90** | **<0.001** | 1.01 | 0.86-1.19 | 0.915 |
| Most deprived | **1.61** | **1.27-2.03** | **<0.001** | 1.02 | 0.84-1.23 | 0.876 |
| **Marital status** | | | | | | |
| Married and living with partner | Ref. | – | – | Ref. | – | – |
| Not married and living with partner | 1.03 | 0.86-1.23 | 0.762 | 0.91 | 0.78-1.06 | 0.211 |
| Not living with partner | **1.19** | **0.99-1.43** | **0.071** | **0.79** | **0.68-0.93** | **0.005** |
| **IMD Quintiles**\*** | | | | | | |
| 1 | Ref. | – | – | Ref. | – | – |
| 2 | 0.98 | 0.82-1.16 | 0.777 | 0.98 | 0.85-1.13 | 0.800 |
| 3 | 0.86 | 0.70-1.04 | 0.117 | 1.02 | 0.87-1.19 | 0.803 |
| 4 | 0.84 | 0.67-1.04 | 0.115 | 1.00 | 0.84-1.20 | 0.958 |
| 5 | **0.60** | **0.37-0.96** | **0.034** | **1.43** | **1.01-2.01** | **0.043** |
| **Parity** | | | | | | |
| Nulliparous | Ref. | – | – | Ref. | – | – |
| 1 | **1.19** | **1.01-1.41** | **0.038** | **0.87** | **0.76-1.00** | **0.045** |
| 2 | **1.29** | **1.05-1.57** | **0.013** | 0.90 | 0.76-1.06 | 0.195 |
| 3+ | **1.42** | **1.13-1.79** | **0.003** | **1.21** | **1.00-1.46** | **0.048** |

*(Continued)*

**Table 2.** (Continued)

| Potential determinants | Less than the RWG | | | More than the RWG | | |
|---|---|---|---|---|---|---|
| | OR | 95% CI | p-value* | OR | 95% CI | p-value* |
| **Smoked during pregnancy** | | | | | | |
| Yes | **1.23** | **1.02-1.47** | **0.027** | **0.87** | **0.75-1.02** | **0.084** |
| No | Ref. | – | – | Ref. | – | – |
| **Previous hypertension** | | | | | | |
| Yes | 1.11 | 0.48-2.57 | 0.802 | 1.44 | 0.73-2.83 | 0.294 |
| No | Ref. | – | – | Ref. | – | – |
| **Baseline BMI††** | | | | | | |
| Underweight | **1.77** | **1.36-2.30** | **<0.001** | **0.59** | **0.44-0.80** | **0.001** |
| Healthy weight | Ref. | – | – | Ref. | – | – |
| Overweight | **1.39** | **1.17-1.64** | **<0.001** | **3.31** | **2.88-3.80** | **<0.001** |
| Obese | **1.38** | **1.13-1.68** | **0.001** | **5.22** | **4.44-6.12** | **<0.001** |

OR: Odds ratio;

*p-values refer to individual Wald tests

**GHQ-28: General health questionnaire. > 75th centile means higher risk of psychiatric morbidity.

***IMD: index of multiple deprivation.

†Categories of SEP (socioeconomic position): "least socioeconomically deprived and most educated"; "employed and not materially deprived"; "employed and no access to money"; "benefits but coping" and "most economically deprived"

††BMI (body mass index) categories' thresholds vary by ethnicity (S2 Table). Adequate weight gain used as reference.

of overweight and obesity on the odds of gaining less than the RWG was also notable in this group: despite wider confidence intervals, there were apparently larger effect sizes when compared to Pakistani participants. There seemed to be a possible protective effect of advanced maternal age (35+) against having more than the RWG among Pakistani women. This occurrence was not identified among White British women. GHQ-28 (mental health) seemed to have a relevant impact in the odds of acquiring higher than recommended weight mainly among Pakistani women.

### Sensitivity analyses

In general, the sensitivity analyses did not lead to major changes in the interpretation of findings (S3–S8 Figs). Nevertheless, when IOM categories for overall absolute weight gain were used as the outcome, requiring restriction of the analysis for the subset of 3,271 participants with late third trimester weight measurement, there was a change in the direction of the association between overweight and obesity and gaining 'less than the RWG' (S5 Fig).

Overall, the results from the complete case analysis (S8 Fig) were consistent with findings from the imputed model (Fig 1).

### Discussion

In this cohort of pregnant women from a multi-ethnic UK population, 22.4% of women gained weight within the 2009 Institute of Medicine recommended range, 20.3% gained less weight than recommended, and 57.3% gained more weight than recommended. Five of the twelve potential determinants under investigation were found to be independently associated with gaining gestational weight below and/or above the IOM guidelines: baseline BMI, parity, socioeconomic position, age and mental health. Lower socioeconomic position, higher parity, unhealthy BMI (underweight, overweight or obese) and young maternal age were associated with higher odds of experiencing less than the recommended GWG. Meanwhile, being overweight, obese or having poor mental health were determinants of having 'more than the RWG', whereas multiparity and underweight were associated with a lower risk of gaining more weight than the recommendations.

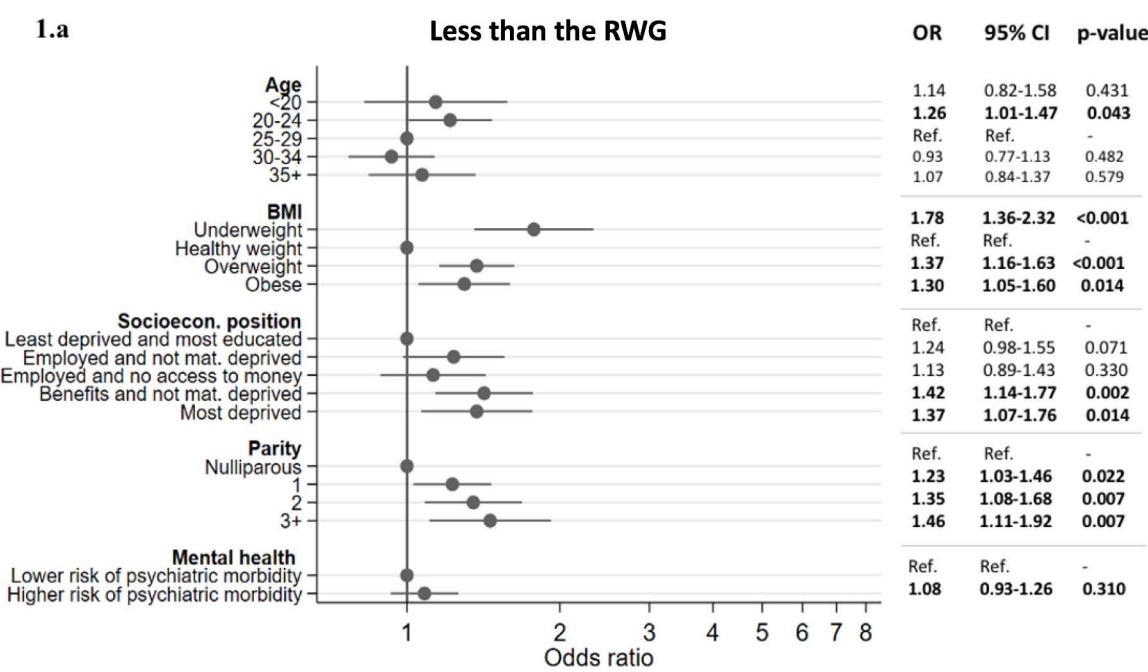

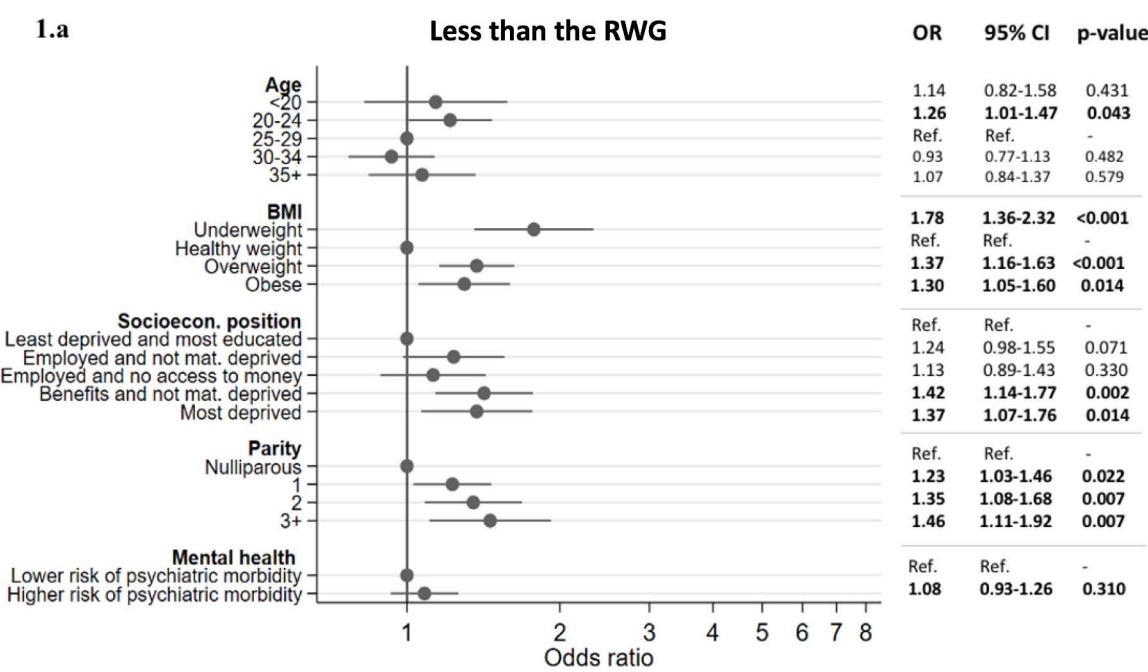

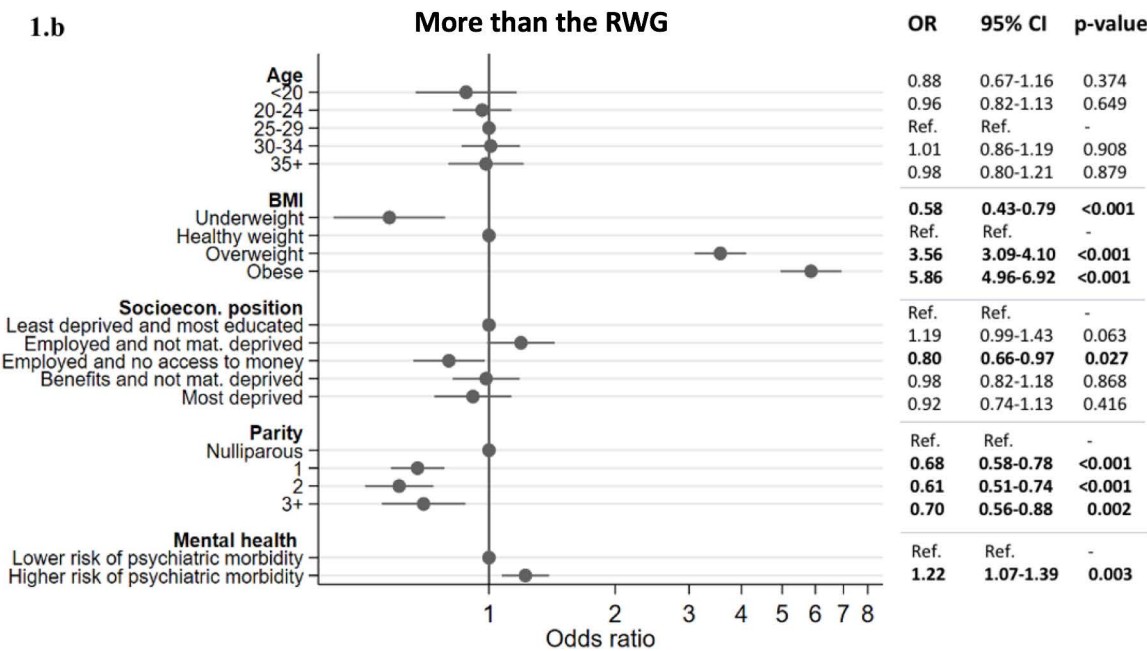

**Fig 1. Plot of determinants of 'less than the recommended weight gain (RWG)' and 'more than the recommended weight gain (RWG)', adjusted for maternal age and ethnicity.**

In accordance with previous studies [9,26], baseline BMI was a consistent and important determinant of a GWG above the recommended range. In a meta-analysis that categorized overweight and obese women into a single group, the OR for excessive weight within this category was 2.49 (95% CI 0.92–1.21) when compared to those with a healthy BMI [9]. Within our cohort, the effect was even stronger: OR=3.56 for baseline overweight, and OR=5.86 for baseline obesity.

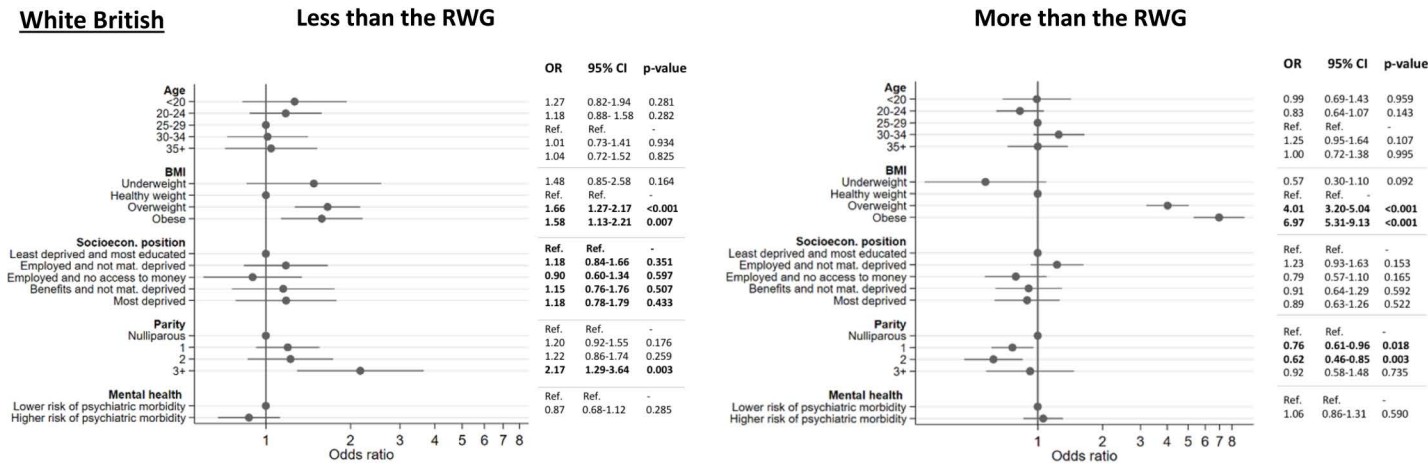

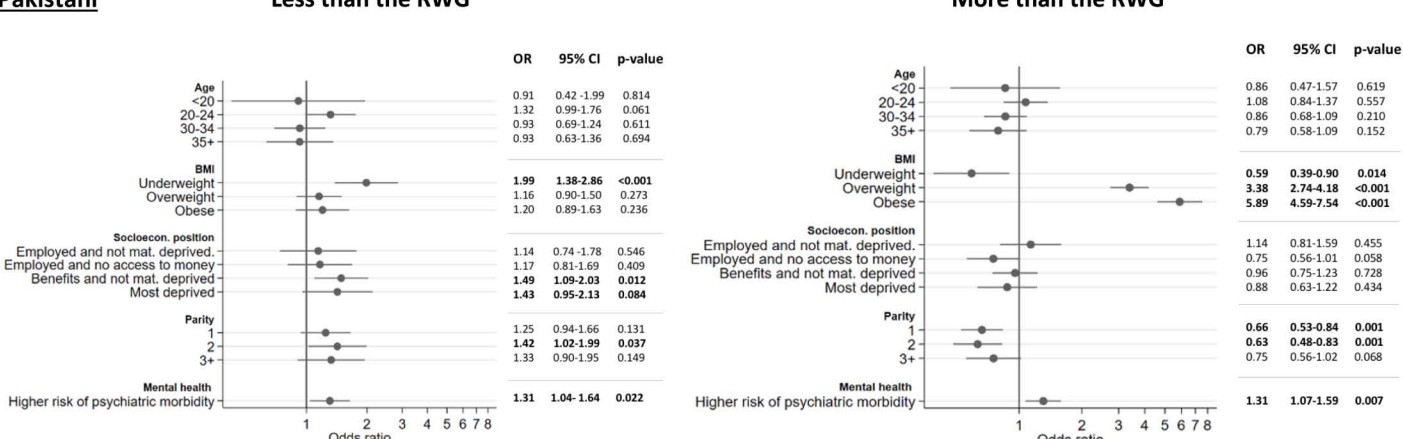

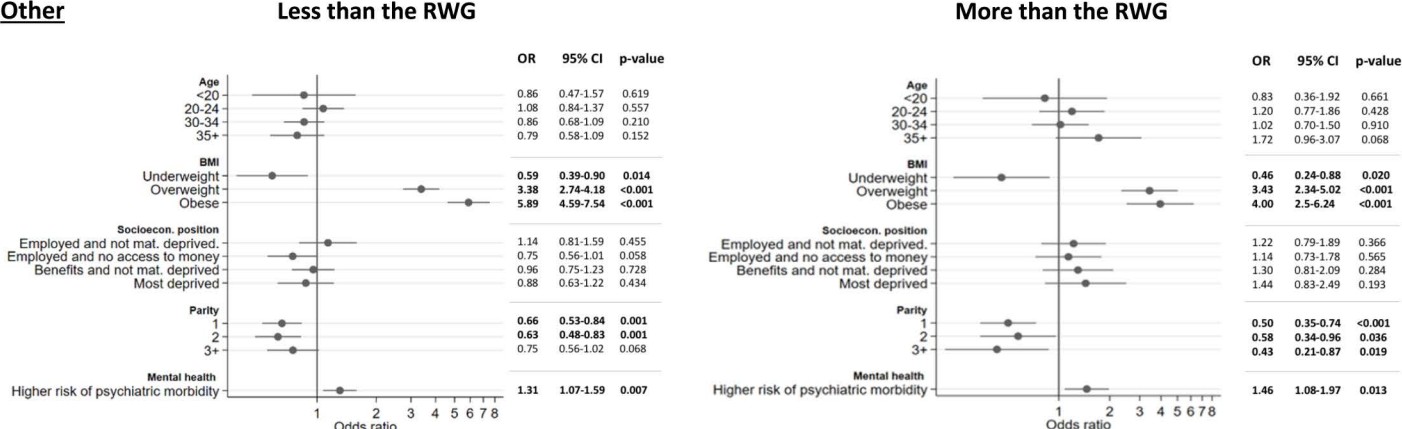

**Fig 2. Subgroup analysis: determinants of 'less than the recommended weight gain (RWG)' and 'more than the recommended (RWG)', adjusted for maternal age and stratified by ethnicity.**

In the same meta-analysis [9], as in the present study, having an underweight baseline BMI nearly halved the odds of experiencing higher than recommended GWG. The mechanisms underlying the relationship between BMI and gestational weight gain above the recommended range are probably multifactorial, encompassing biological and behavioral aspects [27,28].

On the other hand, the nature of the relationship between BMI and 'less than the recommended RWG' is more heterogeneous in the literature. Some studies reported higher odds of experiencing a GWG below the recommended range among obese [26,29] and underweight [29] women, and lower odds among overweight participants [29,30]. However, others failed to detect any association between BMI and less-than-recommended GWG [31]. In our study, all categories of 'unhealthy' baseline BMI were associated with higher odds of acquiring less weight than the reference range when categories of average weekly weight gain were used as the outcome. However, baseline overweight or obesity were associated with decreased odds of gaining less than the RWG in a sensitivity analysis using overall weight gain as the outcome (S5 Fig). The difference in findings did not seem to be due to selection bias, since an analysis of the weekly rate of weight gain restricted to the 3,271 women in this subgroup yielded similar results as the main analysis involving the total sample (results not shown). Possible reasons for the different results may include a lack of a perfect correlation between total pregnancy weight gain and weekly average weight gain [32], the possibility of a non-linear weight gain throughout the second and third trimesters [33], and/or weight gain during the first trimester outside the 0.5–2 kg range assumed by IOM guidelines [19].

The relationship between mental health morbidity and higher odds of having 'more than the RWG' identified in the present study supports the literature. A greater risk of GWG above recommendations among participants with anxiety [34] or depression [13,28] has been described. Interestingly, in the present study, the influence of impaired mental health on GWG seemed to be particularly important among women of non-white British ethnic origin. This opens space for future studies addressing this topic in South Asian populations, migrants and ethnic minorities, which are consistently underrepresented in studies on this subject. Although various studies also identify a link between mental health morbidity and less-than-recommended weight gain [26,35,36], such an association was not found in the present study.

With regard to socioeconomic status, we noted a relevant impact of economic hardship on the odds of gaining less than the recommended GWG. We believe this relationship could be at least partially explained by food insecurity [37,38]. Although two systematic reviews suggested that lower socioeconomic status may also be associated with higher than recommended GWG [37,38], a phenomenon that could be linked to high intake of calorie-dense unhealthy products [39] among those with suboptimal economic position, this relationship was not consistently noted in our study.

In the present investigation, multiparity was associated with a higher risk of gaining 'less than the RWG' and a lower risk of gaining 'more than the RWG'. These findings are consistent with a meta-analysis of 25 studies from middle and low-income countries [12] but differ from a meta-analysis of more than 70 studies [9] that concluded that multiparous women were at higher risk of excessive GWG and from another meta-analysis that failed to find a significant association [40]. When exploring the association between parity and GWG, Hill et al [40] identified remarkable levels of heterogeneity across studies. The authors claim that the relationship is probably 'indirect and complex' [40]. Further studies exploring the potential mechanisms/rationale are warranted, including an investigation of the role of inter-pregnancies intervals and inter-pregnancy weight changes.

Results from the sensitivity analyses yielded similar results to the main analysis, with the exception of the exploring overall weight gain, rather than average weekly weight gain as the outcome.

## Strengths and limitations

Our study has a number of important strengths. It included more than seven thousand participants, which is a large sample size when compared to similar studies. Moreover, we investigated a wide variety of potential determinants of gestational weight gain, from classical to novel factors (such as psychiatric morbidity) [26]. Additionally, there was a great effort from the

BiB cohort to include women from diverse ethnicities, including the administration of questionnaires in various languages [17]. This is a substantial advantage in contrast to other studies, which frequently exclude non-English speakers, even when performed in multi-ethnic populations [26], leading to an under-representation of ethnic 'minorities' in most investigations.

Additionally, while some studies rely on self-reporting of weight (which has been shown to lead to underestimation of the actual values [34]), our investigation used objective measurements and linkage to maternity notes that included information obtained from antenatal records. This is especially important considering that body weight is vulnerable to social desirability bias [41].

However, some methodological limitations must be acknowledged. Obtaining an accurate assessment of gestational weight gain is challenging [42]. Such limitations arise from the differential timing (gestational week) at measurement and a lack of universal pre-pregnancy weight ascertainment. As pre-pregnancy weight is seldom available for the vast majority of participants and the amount of weight gain during the first trimester is small (0.5 to 2 kg) [19], many studies - including the present one - rely on weight at the first antenatal appointment as a proxy for pre-pregnancy weight [42]. This approach can lead to some level of misclassification of baseline BMI. However, as this would mostly impact the categorization of BMI among participants with values close to the category thresholds, the impact of misclassification is likely small in this context and is unlikely to cause major changes in the nature of the associations.

Additionally, the IOM-recommended criteria, although widely used in the literature, might not be optimal for non-US populations, and does not originally account for ethnic differences [19]. As almost half of the participants are from a Pakistani background, we conducted a sensitivity analysis using different cut-offs for BMI categories and a subgroup analysis by ethnicity in an attempt to overcome this limitation.

In addition to the under-representation of individuals from diverse ethnical/racial backgrounds, the IOM guidelines have other inherent limitations. For example, recommendations do not distinguish between class I, II and III obesity. The 2009 guidelines state that available evidence was "insufficient to develop more specific recommendations for GWG" among women with class II and III obesity [19]. Considering the significant prevalence of severe obesity among women of child-bearing age and the growing number of studies investigating the impact of GWG on pregnancy outcomes within different obesity strata, thoroughly readdressing the current evidence base in search of optimal GWG ranges stratified by obesity class would be of value. It is also worth noting that the IOM guidelines were primarily based on observational data [19], thus limiting the assessment of causal effects between GWG and pregnancy outcomes. Additionally, there are knowledge gaps about how ideal weight gain may vary not only by pre-pregnancy BMI but also in relation to other characteristics such as maternal age, height or comorbidities [19]. Despite these limitations, a decision was made to use the 2009 IOM guidelines in the present study, as they are thoughtfully developed, widely adopted in both the literature and medical practice, and enable comparisons across various studies.

Since information on late-third trimester weight was not available for more than half of the sample, we were unable to calculate absolute weight gain for many participants and relied on the rate of weight gain in the main analysis. However, for some pregnant women, the average weekly weight gain might not reflect the rate of weight gain across the whole pregnancy, due to fluctuations at different stages of development. We partially addressed this issue by selecting participants with at least 8 weeks between the initial and final weight measurement and by conducting a sensitivity analysis in the subgroup of women with available late-third trimester data (see sensitivity analysis 3 in supplementary information).

## Conclusions

We believe the present study is an important addition to the literature. In the UK, the National Institute for Health and Care Excellence (NICE) suggested that most women should not be repeatedly weighed during pregnancy [43]. However, Garay et al claim that "women in the UK may benefit from a revised approach towards GWG in the various levels of the NHS" [13]. Our results may help inform organisations on which groups of women are at greater risk of unhealthy weight gain during pregnancy, such as those with mental health issues, unhealthy baseline BMI, and/or facing economic deprivation.

These individuals could be prioritized on intervention programs and receive tailored counseling and follow-up of weight gain during pregnancy as part of their antenatal care.

Finally, the heterogeneity in findings across different studies indicates there may be population-level particularities that mediate the relationships, and this would be an interesting topic for future studies. Experimental studies assessing the potential impact of managing mental health morbidity and promoting a healthy BMI [2] prior to pregnancy could pave the way for better pregnancy outcomes for women and their children.

## Supporting information

**S1 Fig. Participant flow chart.**
(TIF)

**S2 Fig. Plot of determinants of "less than the RWG" (S2.a) and "more than the RWG" (S2.b), adjusted for maternal age and ethnicity, and excluding GHQ-28 score.**
(TIF)

**S3 Fig. Sensitivity analysis 1: determinants of "less than the RWG" (S3.a) and "more than the RWG" (S3.b), excluding women with gestational diabetes.** Results adjusted for ethnicity.
(TIF)

**S4 Fig. Sensitivity analysis 2: determinants of "less than the RWG" (S4.a) and "more than the RWG" (S4.b), when restricting analysis to those with post-recruitment weight gain only.** Results adjusted for ethnicity.
(TIF)

**S5 Fig. Sensitivity analysis 3: determinants of "less than the RWG" (S5.a) and "more than the RWG" (S5.b), in women with absolute (total) weight gain.** Results adjusted for ethnicity.
(TIF)

**S6 Fig. Sensitivity analysis 4: determinants of "less than the RWG" (S6.a) and "more than the RWG" (S6.b), using standard ('Western') BMI cut-offs for all participants.** Results adjusted for ethnicity.
(TIF)

**S7 Fig. Sensitivity analysis 5: determinants of "less than the RWG" and "more than the RWG" whilst correcting for potential first trimester weight gain.** Results adjusted for ethnicity. *S7.a corresponds to the addition of 0.5 kg to weight at first antenatal appointment and setting of gestational age for 14 weeks for participants with first antenatal appointment before 8 weeks. S7.b corresponds to the addition of 2 kg to weight at first antenatal appointment and setting of gestational age for 14 weeks for participants with first antenatal appointment before 8 weeks.*
(TIF)

**S8 Fig. Sensitivity analysis 6: determinants of "less than the RWG" and "more than the RWG" (a complete case analysis).** Results adjusted for ethnicity. *BMI (body mass index) categories' thresholds vary by ethnicity (S2 Table). GHQ-28 was not included in the complete case analysis due to more than 20% of missing data.*
(TIF)

**S1 Table. Institute of Medicine 2009 criteria for recommended gestational weight gain.**
(DOCX)

**S2 Table. BMI cut-offs used for classifying pre-pregnancy BMI according to ethnic background.**
(DOCX)

**S3 Table. Characteristics of participants included vs excluded from the complete case analysis.**
(DOCX)

**S1 File. Supplementary methods.**
(DOCX)

## Acknowledgments

Born in Bradford is only possible because of the enthusiasm and commitment of the children and parents in BiB. We are grateful to all the participants, health professionals, schools and researchers who have made Born in Bradford happen. In addition to this, we gratefully acknowledge the contribution of TPP and the TPP ResearchOne team in completing study participant matching to GP primary care records and in providing ongoing informatics support.

## Author contributions

**Conceptualization:** Maria A. Quigley, Victoria Coathup.

**Data curation:** Gillian Santorelli, Victoria Coathup.

**Formal analysis:** Petra A. T. Araújo.

**Funding acquisition:** Victoria Coathup.

**Methodology:** Maria A. Quigley, Victoria Coathup.

**Writing – original draft:** Petra A. T. Araújo.

**Writing – review & editing:** Maria A. Quigley, Gillian Santorelli, Victoria Coathup.

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
