## [Decision Letter · Decision Letter 0]

21 Nov 2024

PONE-D-24-19375Determinants of gestational weight gain in a multiethnic UK-based population:Findings from the Born in Bradford cohort study.PLOS ONE

Dear Dr. Coathup,

Thank you for submitting your manuscript to PLOS ONE. After careful consideration, we feel that it has merit but does not fully meet PLOS ONE’s publication criteria as it currently stands. Therefore, we invite you to submit a revised version of the manuscript that addresses the points raised during the review process.

We look forward to receiving your revised manuscript.

Kind regards,

Surangi Jayakody, MBBS, MSc, MD

Academic Editor

PLOS ONE

Reviewers' comments:

Reviewer's Responses to Questions

**Comments to the Author**

1. Is the manuscript technically sound, and do the data support the conclusions?

Reviewer #1: Yes

Reviewer #2: Yes

2. Has the statistical analysis been performed appropriately and rigorously? 

Reviewer #1: Yes

Reviewer #2: I Don't Know

3. Have the authors made all data underlying the findings in their manuscript fully available?

Reviewer #1: Yes

Reviewer #2: Yes

4. Is the manuscript presented in an intelligible fashion and written in standard English?

Reviewer #1: Yes

Reviewer #2: Yes

5. Review Comments to the Author

Reviewer #1: This is a good population-based study,all ares of reseach are done properly, the objective of the the study is as very important one as it reflects the the outcome of pregnancy in both maternal and fetal aspects in different ethnic groups living in UK

Reviewer #2: I read this study with much interest and wish to congratulate the authors for addressing a research gap, especially exploring the multi ethnic cohort and including many more determinants than the previous studies. The study certainly deserves dissemination to Plos one readers and the scientific community.

I have a few comments to make.

Use of IOM guidelines is inevitable as this is the best available reference on GWG. Although IOM has strong evidence on “normal weight” followup and evidence base, only moderate evidence base is used for under and overweight/ obese groups. As authors themselves stated, IOM may not be externally valid outside of a USA population.

Therefore, I suggest the use of the term "recommended weight gain (RWG)" instead of “adequate weight gain”, and describe the populations with terms "less than the RWG"(instead of “insufficient weight gain”) and "more than the RWG" (instead of excessive weight gain) to reflect the fact that there is currently no universally agreed "adequate weight gain".

Consider adding a brief discussion of the limitations of IOM guidelines

I note that the study population had "universal screening for gestational diabetes with 75g OGTT". Although this is the recommended current practice, UK in 2007 was using "two step screening" (?). If the BiB population was considered to have a different guideline to be adopted locally, a brief discussion of this fact in the limitations would be of value.

6. PLOS authors have the option to publish the peer review history of their article (what does this mean? ). If published, this will include your full peer review and any attached files.

**Do you want your identity to be public for this peer review?** For information about this choice, including consent withdrawal, please see our Privacy Policy .

Reviewer #1: **Yes: ** Mohsen M A Abdelhafez

Reviewer #2: **Yes: ** Indu Asanka Jayawardane

---

## [Author Response · Author response to Decision Letter 1]

7 Feb 2025

Dear Editor and Reviewers,

Thank you for your time and thoughtful analysis of the manuscript. We greatly appreciate your insightful feedback, and we are confident that implementing your suggestions will strengthen the quality of the work.

As requested, we are submitting a revised version of the manuscript (including a marked-up version with track changes and an unmarked version). Additionally, we are uploading marked and unmarked revised versions of the ‘supporting information’ file and a revised version of figures 1 and 2.

Below, we outline how each of the suggestions and comments has been addressed in the revised version of the manuscript.

1) “I suggest the use of the term "recommended weight gain (RWG)" instead of “adequate weight gain”, and describe the populations with terms "less than the RWG"(instead of “insufficient weight gain”) and "more than the RWG" (instead of excessive weight gain) to reflect the fact that there is currently no universally agreed "adequate weight gain".

a. We appreciate this important advice, and have removed the terms “insufficient”, “adequate” and “excessive” weight gain when referring to our participants throughout the main manuscript and supporting information. These terms have been replaced with "less than the RWG", "recommended weight gain (RWG)", and "more than the RWG" as well as other similar expressions or synonyms, in order to align with sentence structure and syntax, ensure clarity and avoid repetition. Alternative terms used include “GWG below/above the recommended range” and “higher than/lower than recommended”, for example. Figures 1 and 2 (as well as the supplementary figures within the supporting file) have been edited to change the headings from “insufficient weight gain” and “excessive weight gain” to “less than the RWG” and “more than the RWG”.

Please note that there are two sentences where the term “excessive” appears in the manuscript, but specifically when referring to the results of other studies that have employed this nomenclature (lines 484 and 554).

2) “Consider adding a brief discussion of the limitations of IOM guidelines”

a. We approached this recommendation by including a paragraph in the Discussion section, under the ‘Strengths and limitations’ subheading (lines 601 to 615).

“In addition to the under-representation of individuals from diverse ethnical/racial backgrounds, the IOM guidelines have other inherent limitations. For example, recommendations do not distinguish between class I, II and III obesity. The 2009 guidelines state that available evidence was “insufficient to develop more specific recommendations for GWG” among women with class II and III obesity [19]. Considering the significant prevalence of severe obesity among women of childbearing age and the growing number of studies investigating the impact of GWG on pregnancy outcomes within different obesity strata, thoroughly readdressing the current evidence base in search of optimal GWG ranges stratified by obesity class would be of value. It is also worth noting that the IOM guidelines were primarily based on observational data [19], thus limiting the assessment of causal effects between GWG and pregnancy outcomes. Additionally, there are knowledge gaps about how ideal weight gain may vary not only by pre-pregnancy BMI but also in relation to other characteristics such as maternal age, height or comorbidities [19]. Despite these limitations, a decision was made to use the 2009 IOM guidelines in the present study, as they are thoughtfully developed, widely adopted in both the literature and medical practice, and enable comparisons across various studies”.

We hope that this additional paragraph, along with the one precedes it (lines 596 to 600), helps to highlight key concerns related to the IOM guidelines.

3) I note that the study population had "universal screening for gestational diabetes with 75g OGTT". Although this is the recommended current practice, UK in 2007 was using "two step screening" (?). If the BiB population was considered to have a different guideline to be adopted locally, a brief discussion of this fact in the limitations would be of value.

The BiB protocol for the recruitment phase (available on https://bmcpublichealth.biomedcentral.com/articles/10.1186/1471-2458-8-327) states that “Bradford is served by a single maternity unit, at the Bradford Royal Infirmary (BRI)” and that “almost all women resident in Bradford book, and give birth, in the maternity unit”. “All women booked for delivery at the BRI are offered an oral glucose tolerance test (oral GTT) at 26 to 28 weeks gestation. When they attend for the oral GTT, women are invited to participate in the Born in Bradford cohort”.

Therefore, at the time of the study recruitment, oral GTT was being offered universally for pregnant women in the city of Bradford. We have revised the sentences regarding this topic in the methods section, to provide clearer explanation. This now reads as follows (lines 151 to 154):

“Most women were typically enrolled at 26-28 weeks of gestational age when they attended an ‘oral glucose tolerance test’ (oral GTT) appointment. The oral GTT was offered to all pregnant women booked for delivery at the Bradford Royal Infirmary, with an uptake of approximately 80% [18]”.

We believe that offering the oral GTT to all pregnant women in Bradford did not negatively impact the results of our study neither led to bias. We feel that it provided an opportune moment to recruit a large number of potential participants, improving the representativeness of the study sample. Additionally, it may have enhanced the detection of gestational diabetes mellitus, thereby improving the accuracy of the sensitivity analysis in which women with gestational diabetes were excluded. As a result, instead of addressing this topic in the ‘limitations’ section, we opted to rephrase the excerpt in the methods section, as noted above. We kindly hope this will be an acceptable alternative.

4) In addition to the aforementioned modifications, the following formatting changes have been made:

a. We adjusted the font sizes in the manuscript body as follows: 18pt for Level 1 headings, 16pt for Level 2 headings, and 14pt for Level 3 headings. Additionally, we replaced headings in italics with bold type, in accordance with the PLOS ONE style templates.

b. The "References" section has been moved to appear before the supporting information, in accordance with the PLOS ONE style templates.

We would like to sincerely thank you once again for your thoughtful comments and constructive feedback. We hope that the revisions we have made address your concerns effectively.

---

## [Decision Letter · Decision Letter 1]

6 Apr 2025

Determinants of gestational weight gain during pregnancy in a multiethnic UK-based population: Findings from the Born in Bradford cohort study

PONE-D-24-19375R1

Dear Dr. Coathup,

We’re pleased to inform you that your manuscript has been judged scientifically suitable for publication and will be formally accepted for publication once it meets all outstanding technical requirements.

Kind regards,

Guoying Wang, MD, PhD

Academic Editor

PLOS ONE

Additional Editor Comments (optional):

Reviewers' comments:

Reviewer's Responses to Questions

**Comments to the Author**

1. If the authors have adequately addressed your comments raised in a previous round of review and you feel that this manuscript is now acceptable for publication, you may indicate that here to bypass the “Comments to the Author” section, enter your conflict of interest statement in the “Confidential to Editor” section, and submit your "Accept" recommendation.

Reviewer #1: All comments have been addressed

Reviewer #2: All comments have been addressed

2. Is the manuscript technically sound, and do the data support the conclusions?

Reviewer #1: Yes

Reviewer #2: Yes

3. Has the statistical analysis been performed appropriately and rigorously? 

Reviewer #1: Yes

Reviewer #2: Yes

4. Have the authors made all data underlying the findings in their manuscript fully available?

Reviewer #1: Yes

Reviewer #2: Yes

5. Is the manuscript presented in an intelligible fashion and written in standard English?

Reviewer #1: Yes

Reviewer #2: Yes

6. Review Comments to the Author

Reviewer #1: Thank you for proper addressing all the comments given to authors however, extention of research is still required

Reviewer #2: Thank you for the interesting manuscript. I agrre with the authors in that universal OGTT was a window of opportunity at the time and therefore the amendment is best described at "Method". My concerns have been addressed fully. I feel this is now ready for publication.

7. PLOS authors have the option to publish the peer review history of their article (what does this mean? ). If published, this will include your full peer review and any attached files.

**Do you want your identity to be public for this peer review?** For information about this choice, including consent withdrawal, please see our Privacy Policy .

Reviewer #1: **Yes: ** Mohsen M A Abdelhafez

Reviewer #2: **Yes: ** Indu Asanka Jayawardane

---

## [Editor Report · Acceptance letter]

PONE-D-24-19375R1

PLOS ONE

Dear Dr. Coathup,

I'm pleased to inform you that your manuscript has been deemed suitable for publication in PLOS ONE. Congratulations! Your manuscript is now being handed over to our production team.

Kind regards,

on behalf of

Dr. Guoying Wang

Academic Editor

PLOS ONE